TOPICAL REVIEW

# Corticomotor pathway function and recovery after stroke: a look back and a way forward

Maxine J. Shanks[1,2] ⓘ and Winston D. Byblow[1,2] ⓘ

[1]*Department of Exercise Sciences, University of Auckland, Auckland, New Zealand*
[2]*Centre for Brain Research, University of Auckland, Auckland, New Zealand*

Handling Editors: Laura Bennet & Ricci Hannah

The peer review history is available in the Supporting Information section of this article (https://doi.org/10.1113/JP285562#support-information-section).

**Abstract** Stroke is a leading cause of adult disability that results in motor deficits and reduced independence. Regaining independence relies on motor recovery, particularly regaining function of the hand and arm. This review presents evidence from human studies that have used trans-cranial magnetic stimulation (TMS) to identify neurophysiological mechanisms underlying upper limb motor recovery early after stroke. TMS studies undertaken at the subacute stage after stroke have identified several neurophysiological factors that can drive motor impairment, including membrane excitability, the recruitment of corticomotor neurons, and glutamatergic and

**Maxine J. Shanks** has a BSc (Hons) in Neuroscience and a BPhty (Hons) and is currently completing her PhD in movement neuroscience. Additionally, she is a registered physiotherapist specializing in neuro-rehabilitation. Her thesis is exploring the neurophysiology of upper limb motor recovery after stroke.
**Winston D. Byblow** is a neurophysiologist and Professor of Neuroscience at the University of Auckland, where he is Director of the Movement Neuroscience Laboratory. Researchers in his laboratory investigate the functions of the nervous system that pertain to sensorimotor control and the cognitive control of action, and neural mechanisms underlying movement disorders and neurological recovery, especially after stroke.

GABAergic neurotransmission. However, the inherent variability and subsequent poor reliability of measures derived from motor evoked potentials (MEPs) limit the use of TMS for prognosis at the individual patient level. Currently, prediction tools that provide the most accurate information about upper limb motor outcomes for individual patients early after stroke combine clinical measures with a simple neurophysiological biomarker based on MEP presence or absence, i.e. MEP status. Here, we propose a new compositional framework to examine MEPs across several upper limb muscles within a threshold matrix. The matrix can provide a more comprehensive view of corticomotor function and recovery after stroke by quantifying the evolution of sub-threshold and suprathreshold MEPs through compositional analyses. Our contention is that subthreshold responses might be the most sensitive to reduced output of corticomotor neurons, desynchronized firing of the remaining neurons, and myelination processes that occur early after stroke. Quantifying subthreshold responses might provide new insights into post-stroke neurophysiology and improve the accuracy of prediction of upper limb motor outcomes.

(Received 26 December 2023; accepted after revision 15 May 2024; first published online 31 May 2024)

**Corresponding author** W. Byblow: Department of Exercise Sciences, University of Auckland, Private Bag 92019, Auckland 1142, New Zealand.    Email: w.byblow@auckland.ac.nz

**Abstract figure legend** Monohemispheric stroke damages cortical neurons, resulting in cell death or demyelination. Consequently, descending output is desynchronized, and motor evoked potentials (MEPs) elicited by transcranial magnetic stimulation (TMS) can be small and polyphasic. As a result, the slope of the stimulus–response (S-R) curve can be shallower than normal (left side of diagram). A threshold matrix depicts responses from multiple upper limb muscles and TMS intensities to capture the small, subthreshold responses more accurately after stroke. Spontaneous biological recovery mechanisms are at play and involve processes that include remyelination of axons in surviving neurons. As a result, more synchronized motor output is recovered, the frequency of subthreshold MEPs decreases, and the slope of the S-R curve increases (top right). Irreversible neuronal damage beyond a point of no return can prevent this pattern of recovery, resulting in persistent subthreshold MEPs as attempts at remyelination are insufficient to produce a synchronized motor output (bottom right).

## Introduction

Stroke is a leading cause of mortality and disability (Johnson et al., 2019). After stroke, motor deficits are common (Langhorne et al., 2009). Regaining independence after stroke relies heavily on motor recovery, particularly recovery of upper limb activity (Stinear et al., 2020; Veerbeek et al., 2011). This topical review focuses on synthesizing what is known about the neurophysiological basis of upper limb motor recovery after stroke. The review focuses on transcranial magnetic stimulation (TMS) studies during the post-stroke sub-acute stage, when spontaneous biological processes occur and patients make most of their functional recovery. Neuroimaging and electroencephalographic investigation are outside the scope of this review.

Making accurate predictions about hand and arm functional outcomes for individual patients can be difficult (Stinear, 2010). For example, predictions made by experienced clinicians about upper limb outcomes at 6 months can be no better than chance (Nijland et al., 2013). This is because patients with similar initial upper limb motor impairment can achieve a wide range of functional outcomes. Despite the challenges, prediction

is important because it provides objective information about an individual's recovery potential, which can be used to set realistic goals during rehabilitation and subsequent discharge planning (Stinear, Byblow, Ackerley, Barber, et al., 2017). Neurophysiological techniques can help to distinguish between patients who will make a good versus poor upper limb recovery based on their corticomotor pathway function.

Transcranial magnetic stimulation can be used to investigate corticomotor pathway function during post-stroke motor recovery (Beaulieu & Milot, 2018; Bembenek et al., 2012; Escudero et al., 1998; Heald et al., 1993; Hendricks et al., 2002; Talelli et al., 2006). In recent years, motor evoked potential (MEP) status has been identified as a biomarker for upper limb recovery after stroke (Boyd et al., 2017; Byblow et al., 2015; Stinear, Byblow, Ackerley, Smith, et al., 2017). For example, within 7 days of stroke, patients who are MEP$^+$ tend to experience better functional outcomes than those who are MEP$^-$. Currently, the most accurate tool for predicting upper limb motor outcomes combines MEP status as a biomarker of corticomotor pathway function with other baseline clinical and demographic measures (Stinear et al., 2012; Stinear, Byblow, Ackerley, Smith,

et al., 2017). This review has two aims. First, it revisits the use of TMS-derived measures used to predict recovery potential and understand post-stroke recovery neurophysiological mechanisms. Second, it introduces a new TMS-derived framework that might have the potential to expand understanding of these areas.

## Transcranial magnetic stimulation-derived measures after stroke

**Using TMS to probe the corticomotor pathway.** It is approaching four decades since Barker introduced TMS as a non-invasive method to stimulate the human primary motor cortex (M1) to elicit MEPs (Barker et al., 1985). Although recording MEPs from EMG of the contralateral limb is technically easy, the contributing neurophysiology is complex and not fully understood. Most TMS studies target hand and forearm muscles, owing to the superficial location of the hand motor cortex and its direct corticomotoneuronal projections to alpha motor neurons in the anterior horn of the spinal cord, in addition to spinal interneurons located in the grey matter of the spinal cord. The current consensus is that TMS elicits MEPs in the target musculature by preferentially depolarizing fast-conducting myelinated axons in the underlying cortex (Rothwell, 1997; Siebner et al., 2022). The excitation spreads through extensive corticocortical and corticosubcortical pathways through both excitatory and inhibitory nodes (Di Lazzaro et al., 1998, 2004). The following sections introduce conventional TMS measures and the underlying physiological mechanisms of each measure and explain how each measure is affected after stroke.

**Single-pulse TMS measures.** Single-pulse TMS is used to evaluate the excitability of the corticomotor pathway in several simple ways. Measures of motor threshold reflect the ease with which the corticomotor pathway is activated by TMS. They can be determined either with the muscle at rest [resting motor threshold (RMT)] or in a pre-activated state [active motor threshold (AMT)]. The conventional criterion for RMT is the lowest stimulation intensity required to elicit an MEP of 50 $\mu$V in $\geq$5 of 10 trials (Rossini et al. 2015). The AMT is defined as the lowest intensity to produce a 200 $\mu$V MEP in $\geq$5 of 10 trials (Vucic et al., 2023). Numerous technical and physiological factors should be kept consistent when determining motor threshold, such as coil position, the level of background activity in the target muscle, and environmental noise (Rossini et al., 2015). Pharmacological studies have shown that voltage-gated sodium channel antagonists increase both RMT and AMT. In contrast, voltage-gated sodium channel agonists lower them, confirming that these measures reflect the membrane excitability of

cortical neurons (Ziemann et al., 2015). The density of corticomotor projections also influences motor threshold, with intrinsic hand muscles having lower thresholds than more proximal muscles at the elbow or shoulder (Chen et al. 1998). After stroke, both RMT and AMT are higher in the ipsilesional than in the contralesional hemisphere (Bütefisch et al., 2008; Byblow et al., 2015; Liepert et al., 2000; Manganotti et al., 2002; Swayne et al., 2008). Ipsilesional RMT decreases during the initial weeks post-stroke but largely stabilizes by 4−6 weeks post-stroke (Byblow et al., 2015; Swayne et al., 2008). Given that motor threshold reflects membrane excitability, an early increase in thresholds can be explained physiologically by the hypoxia-driven disruption of ionic gradients and, subsequently, the membrane potentials of the damaged neurons. Despite some recovery, RMT can remain elevated in comparison to the contralesional hemisphere several months after stroke. While there are dynamic changes in ipsilesional RMT, the contralesional RMT remains relatively stable over time and is no different compared with healthy age-matched controls (Bütefisch et al., 2008; Cirillo et al., 2020; Manganotti et al., 2002; Schambra et al., 2015; Swayne et al., 2008).

Applying single-pulse TMS across a wide range of intensities allows stimulus–response (S-R) curves to be constructed. The curves show a sigmoidal relationship between stimulus intensity and MEP amplitude (Devanne et al., 1997). The slope of the S-R curve reflects the gain of the corticomotor pathway, representing the number of corticomotor neurons recruited with stimulus intensity, the density of corticomotor projections, and glutamatergic neurotransmission (Chen et al., 1998; Devanne et al., 1997; Rossini et al., 2015; Rothwell et al., 1987; Stagg, Bestmann, et al., 2011). At the subacute stage, the slope of the S-R curve has been shown to be sensitive to recovery (Stinear et al., 2014; Swayne et al., 2008). However, other studies have shown no difference in the slope of the S-R curve between the ipsilesional and contralesional hemispheres in subacute stroke patients and healthy control hemispheres (Bütefisch et al., 2008; Schambra et al., 2015). The sensitivity of the S-R curve slope to motor recovery might depend on stroke severity. A sigmoidal relationship might not exist in patients with severe upper limb impairment because MEPs can be small and might not modulate with stimulus intensity. At the other end of the spectrum, the S-R curve slope might lack sensitivity to capture the recovery of patients with mild upper limb impairment because the gain of the corticomotor pathway is high enough to compensate for the deficit.

After stroke, MEPs in the paretic upper limb can also be present when TMS is applied to the contralesional (ipsilateral) motor cortex, usually at high intensities (Turton et al., 1996). These ipsilateral MEPs (iMEPs) are more commonly observed in proximal muscles above the elbow or shoulder and less commonly observed

in distal hand or forearm muscles. They also tend to occur at a longer latency than contralateral MEPs. It was noted that for patients at the chronic stage after stroke, the prevalence of iMEPs was negatively associated with upper limb impairment score, i.e. more prevalent iMEPs are associated with worse impairment (Schwerin et al., 2008). These findings suggest that after stroke, there might be upregulation of the contralesional motor cortex to re-establish previously latent synapses onto the alpha motor neurons via indirect descending pathways (McPherson et al., 2018). Early post-stroke upper limb recovery might be mediated along a previously latent descending pathway between the contralesional motor cortex, the reticular formation of the brainstem and propriospinal neurons (Mazevet et al., 2003). Less is known about whether or how iMEPs are upregulated very early after stroke, and how they relate to the extent of spontaneous early recovery is not well understood (Alagona et al., 2001; Bastings et al., 1997; Hammerbeck et al., 2021). Therefore, the remainder of the review will focus on studies that have examined contralateral MEPs.

**Paired-pulse TMS measures.** Paired-pulse TMS protocols can be used to examine GABA-mediated neuro-transmission. Short-interval intracortical inhibition (SICI) occurs when a subthreshold conditioning stimulus (CS) precedes a suprathreshold test stimulus (TS) by 1−5 ms. The resulting test response is inhibited owing to a combination of axonal refractoriness and post-synaptic inhibition mediated by $GABA_A$ receptors (Fisher et al., 2002; Ilic et al., 2002; Roshan et al., 2003; Ziemann et al., 1996). Physiologically, the M1 intracortical circuits reflected by SICI might facilitate the activation of task-specific muscles while suppressing unwanted muscle activity (Stinear & Byblow, 2003).

Elevated GABA-mediated inhibition can present a barrier to plasticity and motor recovery after stroke. Reducing this excessive inhibition during the sub-acute stage of injury promotes functional recovery (Alia et al., 2016; Clarkson et al., 2010). The role of GABA in regulating plasticity in human M1 is also evident (Ziemann et al., 2001), with a key link between the dynamic range of GABA modulation and motor learning capacity (Floyer-Lea et al., 2006; Stagg, Bachtiar, et al., 2011). Over time, SICI in the contralesional hemisphere appears to normalize and has been suggested to be linked to the functional recovery of the patient (Manganotti et al., 2002; Schambra et al., 2015). Some studies detected increased levels of SICI in the ipsilesional hemisphere in patients at the subacute stage of recovery compared with healthy control subjects (Cirillo et al., 2020). Other studies have found decreased SICI in the ipsilesional hemisphere compared with healthy control subjects (Bütefisch et al., 2008; Manganotti et al., 2002; Schambra et al., 2015).

Using a suprathreshold CS and TS with an interstimulus interval of 50–200 ms results in long-interval intracortical inhibition (LICI), which is a measure of cortical inhibition mediated by $GABA_B$ receptors (McDonnell et al., 2006), with no clear consensus about LICI modulation early after stroke compared with healthy control subjects (Cirillo et al., 2020; Swayne et al., 2008). Although a consensus is lacking, there is a recognition that abnormalities in SICI and LICI early after stroke will influence motor output owing to the role of M1 GABAergic neurons in regulating and shaping corticomotor output.

An interhemispheric competition model was proposed by Murase & colleagues (2004) as an explanation for asymmetric motor cortex excitability after stroke (Boroojerdi et al., 1996; Shimizu et al., 2002). The model suggests that decreased ipsilesional hemisphere excitability as a result of stroke leads to asymmetric interhemispheric inhibition between hemispheres and excessive contralesional excitability, which might impede motor recovery (Duque et al., 2005; Murase et al., 2004; Vucic et al., 2023). However, meta-analyses of neurophysiological effects of stroke across the sub-acute and chronic stages have confirmed that differences from healthy control subjects are seen predominantly in measures obtained from TMS of the lesioned hemi-sphere (McDonnell & Stinear, 2017; Veldema et al., 2021). Differences in excitability between hemispheres at the chronic stage are therefore likely to develop over time as a result of asymmetric limb use, rather than being the underlying neurophysiological mechanism causing poor motor recovery (Schambra et al., 2015; Xu et al., 2019). At present there is little neurophysiological evidence to support the interhemispheric competition model at the subacute phase (Dehno et al., 2022; Ejaz et al., 2018). Despite this lack of evidence, the majority of repetitive TMS studies after stroke undertaken since 2005 have selected an interventional protocol on the basis of pre-sumed interhemispheric competition (Safdar et al., 2023).

**Variability of TMS-derived measures.** Although there are established TMS-derived measures of intra- and intercortical activity, it is worth noting that MEPs are inherently variable. This is because the excitability of neuronal pools, at both cortical and spinal levels, fluctuates (Rossini et al., 2015). The timing of the stimulus in relationship to how close or far away the neuronal membrane potentials are from threshold can influence whether a MEP will be elicited. Likewise, the size of elicited MEPs can depend on the timing of the stimulus in relationship to the peaks and troughs of specific cortical oscillations (Ilmoniemi & Kičić, 2010; Tremblay et al., 2019). Finally, activation of the target muscle will decrease the threshold required to produce a MEP, and the MEP amplitude will be larger than

that of a muscle at rest (Devanne et al., 1997). In addition to the inherent variability of MEPs, conflicting neurophysiological results across studies at the subacute stage are influenced by the heterogeneity of groups and differences in assessment time points and TMS protocols. Schambra et al. (2015) comprehensively measured the reliability of TMS measures in subacute and chronic stroke patients and neurologically healthy control subjects. In all groups, RMT had a relatively low measurement error and excellent reliability, whereas SICI and LICI measures had considerable measurement error and moderate reliability. The smallest detectable change was large when analysing MEP variables at the single-participant level, but substantially less for even small groups, validating the use of MEP variables to quantify physiological effects at the group level. The large smallest detectable change at the individual level might explain why MEP status is the only metric that has prognostic value for predicting upper limb motor outcome.

### Limitations of current TMS-derived measures

Resting motor threshold and MEP status are conventional, robust metrics used at the subacute stage after stroke. However, they are not without limitations. The conventionally accepted definition for RMT is based on both amplitude and persistence criteria (Rossini et al., 2015). However, historical limitations of recording environments have led to an arbitrary amplitude criterion for RMT, which might be too conservative, considering technological advancements. Capturing MEPs that fall below the conventional RMT criteria from both amplitude and persistence aspects would seem possible and even informative. Indeed, it has recently been shown that responses of $<50$ $\mu$V can be recorded consistently and are more towards the middle of the linear portion of the S-R curve than once thought (Li et al., 2022). Given that most TMS investigations stimulate at an intensity above RMT, it is plausible that data representing subthreshold corticomotor pathway activation have been overlooked or discarded. This oversight might limit our understanding of the neurophysiological mechanisms responsible for early recovery after stroke.

The MEP status biomarker is an integral element of the PREP prediction tools. For example, the PREP2 prediction tool can be used to predict upper limb outcomes for individual patients within 1 week of stroke. Patients with upper limb weakness receive TMS to determine their MEP status. If they are MEP$^+$, they are predicted to achieve a good functional outcome for their hand and arm at 12 weeks, although they have little or no voluntary movement at the time of testing. Most of these patients meet or exceed their outcome. However, 20% fall short of their predicted outcome (Stinear, Byblow,

Ackerley, Smith, et al., 2017). The variability in upper limb outcomes achieved by MEP$^+$ patients might be influenced by neurophysiological factors not captured with the present binary MEP status biomarker. Extending the present MEP$^+$ category to differentiate between patients with sub- or suprathreshold MEPs across several functionally relevant upper limb muscles might improve prediction accuracy.

The proportional recovery rule states that on average, MEP$^+$ patients will recover $\sim$70% of their maximum possible improvement by 26 weeks post-stroke (Byblow et al., 2015; Prabhakaran et al., 2008). Conversely, MEP$^-$ patients experience very little or no recovery. For MEP$^+$ patients, over the same time frame, the ipsilesional RMT recovers by a remarkably similar proportion to upper limb motor impairment scores (Byblow et al., 2015). A possible mechanism underlying post-stroke deficits and recovery is demyelination and remyelination processes. Oligodendrocytes, responsible for myelination, are more susceptible than axons to the ischaemia caused by stroke (Mifsud et al., 2014). Corticomotor neurons can lose their myelination and, therefore, conduction ability early post-stroke. As remyelination occurs, a process that has been shown to take $\sim$12 weeks (Kondiles & Horner, 2018), corticomotor axonal conduction is restored, and resolution of RMT and motor impairment follow (Byblow et al., 2015). These findings inform us about neurobiological processes underlying early stroke recovery and remind us about the importance of MEP status for predicting outcome and recovery.

Subthreshold MEPs could advance our understanding of initial damage and recovery of the corticomotor pathway early after stroke at a single-patient level. Subthreshold responses can reflect reduced corticomotor neurons, the desynchronized firing of remaining neurons, and demyelination and remyelination processes. Capturing subthreshold responses early after stroke, when spontaneous biological recovery processes occur, might provide a more sensitive assay of corticomotor pathway impairment, function and recovery. We recently proposed a new TMS-derived threshold matrix framework to examine the peri-threshold activation of the corticomotor pathway (Shanks et al., 2023).

### A threshold matrix provides a compositional analysis of corticomotor function

A new threshold matrix framework allows MEPs elicited from TMS to be examined easily across a range of upper limb muscles and stimulation intensities. A schematic diagram of a generic threshold matrix framework is shown in Fig. 1. In a study of healthy adults aged $\geq$50 years to reflect the typical age of first stroke, Shanks et al. (2023) constructed a threshold matrix from single-pulse

TMS across a range of intensities between 10 and 100% maximum stimulator output (MSO) and by recording from four upper limb muscles: first dorsal interosseous, abductor digiti minimi, extensor carpi radialis and flexor carpi radialis. The resulting threshold matrix delineated between three possible responses: suprathreshold MEPs that meet the conventional criteria for RMT; subthreshold MEPs below the RMT criterion; and subliminal responses, which are the combinations of muscles and intensities with no detectable MEPs. For visualization, a traffic light colour scheme can be used to denote supra-threshold, subthreshold and subliminal elements as green, amber and red, respectively. The threshold matrix can also be quantified precisely using compositional data analysis approaches that are appropriate when elements are constrained to sum to 100% (Greenacre, 2021). It is important to capture the relative proportions of the subliminal, subthreshold and suprathreshold elements. One benefit of compositional data analysis is that the relationship between the elements is retained while removing the dependency between elements.

A schematic diagram of a threshold matrix using the same approach as Shanks et al. (2023) is shown in Fig. 2. The conventional way to visualize a three-part composition is a ternary plot. Each vertex of the triangular plot corresponds to 100% of a distinct element, while the edges delineate a scale illustrating the proportions of each element in a given composition. A ternary plot for the threshold matrix data is shown in Fig. 3. Notice how both visualization approaches can capture and quantify subthreshold responses. Including subthreshold responses

expands the linear continuum between subliminal and suprathreshold responses, establishing a distinctive space on the ternary plot. This approach might provide a more comprehensive understanding of the neurophysiological processes that might be particularly relevant early after stroke, when MEPs are often blunted or absent.

The ternary plot also identifies a potential limitation of binarizing MEP status. The MEP$^-$ patients have a 100% subliminal threshold matrix and are therefore situated at the vertex of the ternary plot, as shown in Fig. 4. The rest of the ternary plot represents MEP$^+$ patients. It is plausible that there are at least two distinct MEP$^+$ phenotypes. It is well known that there is variability in the upper limb functional outcome achieved by MEP$^+$ patients (Stinear et al., 2012; Stinear, Byblow, Ackerley, Smith, et al., 2017). Figure 4 shows a hypothetical example with two distinct areas of the ternary plot that might differentiate between recovery phenotypes. Patients with a threshold matrix composition that falls within the purple ellipse have relatively few subthreshold responses and a matrix similar to a neurologically healthy older adult population (Shanks et al., 2023). These patients might meet or exceed their predicted good upper limb functional outcome. Conversely, threshold matrices that fall within the blue ellipse have larger proportions of subthreshold responses. The MEP$^+$ status and therefore a predicted good outcome might be too optimistic for patients with a threshold matrix within this blue ellipse. These patients might not achieve their predicted good outcome or might need additional therapeutic input to achieve a good outcome. The range of possible threshold matrices might

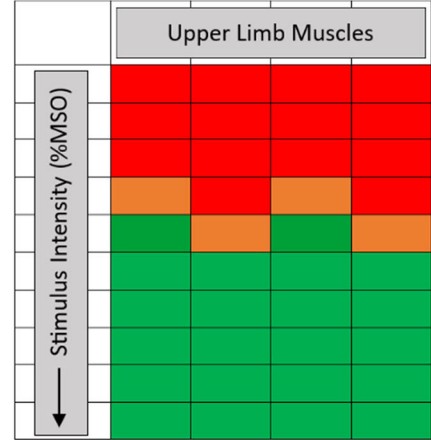
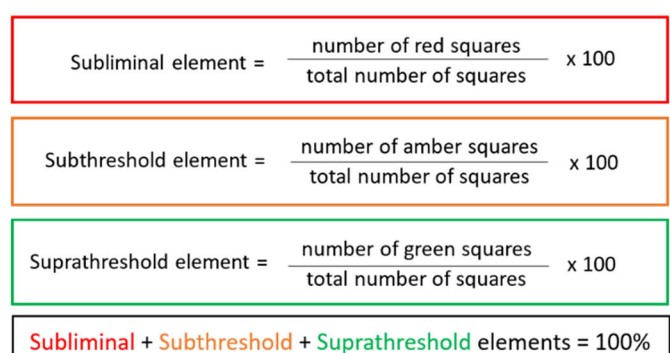

**Figure 1. Threshold matrix construction**
A threshold matrix framework comprises a number of proximate upper limb muscles (columns) and transcranial magnetic stimulation intensities as a percentage of maximum stimulator output (rows). Numerous variations are possible. Cells that produce a set of motor evoked potentials (MEPs) that meet the criteria for resting motor threshold are coloured green. Cells that produce a set of MEPs that fail to meet resting motor threshold criteria are coloured amber. Cells that do not elicit MEPs are coloured red. The threshold matrix composition is calculated by determining the proportion of each coloured element. The red, amber and green elements are termed the subliminal, subthreshold and suprathreshold elements, respectively, and sum to 100%. Abbreviation: %MSO, percentage of maximum stimulator output.

carry important predictive information not captured by the present binary MEP status.

It is worth emphasizing that the discussion of the ternary plot application thus far relates to predicting upper limb motor outcomes within the initial week post-stroke. During this period, the acquired threshold matrix composition offers valuable insight into the present state of the physiology of the corticomotor pathways, thereby aiding in predicting the subsequent outcomes for individual patients. However, this approach does not account for potential early variations in the threshold matrix and movement between different regions of the

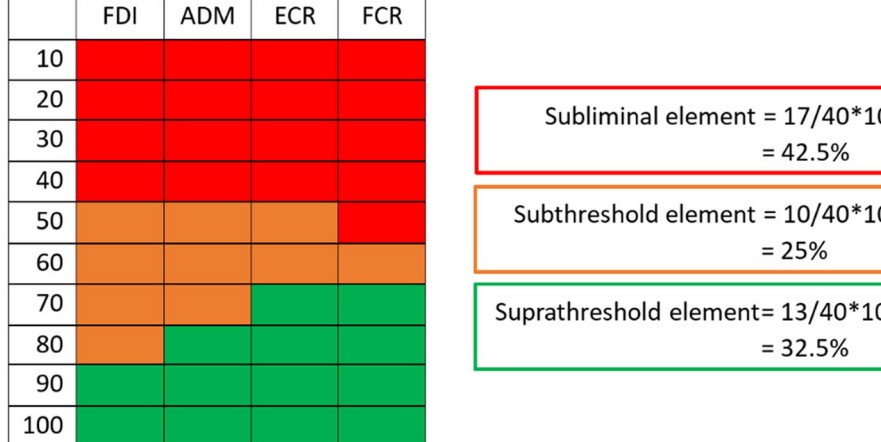

**Figure 2. Threshold matrix from motor evoked potentials obtained from the paretic side of a person 1 month post-stroke with persistent upper limb impairment**
A 40-cell threshold matrix is used, whereby responses are recorded from two intrinsic hand (FDI and ADM) and two forearm (ECR and FCR) muscles. Stimulus intensities range from 10 to 100% in increments of 10% maximum stimulator output. The composition of the matrix is calculated by dividing the number of subliminal, subthreshold and suprathreshold cells by 40 and multiplying by 100. Note the higher thresholds and larger range of stimulus intensities that produce subthreshold motor evoked potentials compared with Fig. 1. Abbreviations: ADM, abductor digiti minimi; ECR, extensor carpi radialis; FCR, flexor carpi radialis; FDI, first dorsal interosseous.

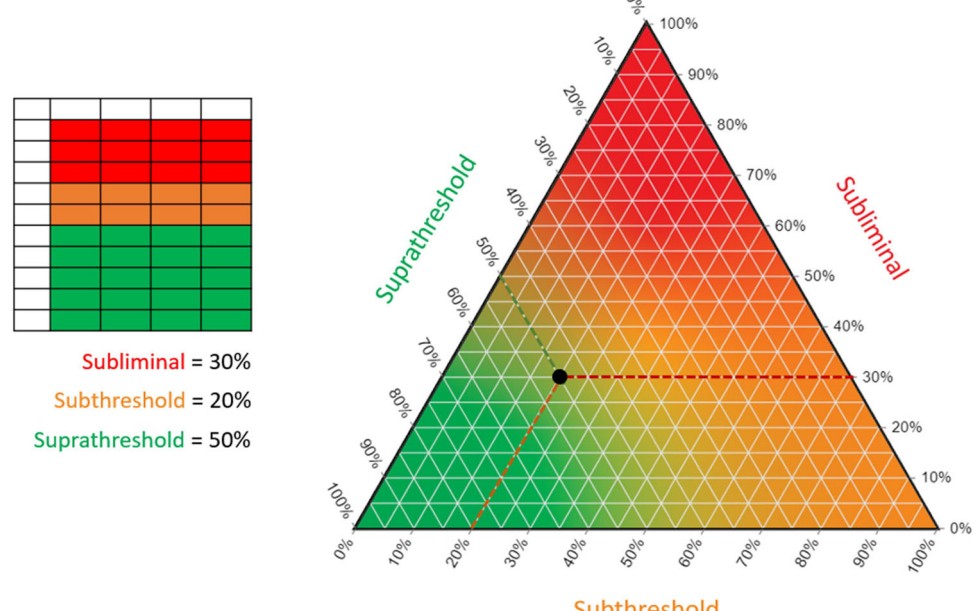

**Figure 3. Visualization with a ternary plot**
Threshold matrix composition (left panel) is depicted as the black dot on a ternary plot (right panel). Each vertex represents one element of the threshold matrix, as the colour map illustrates. The composition is quantified by gridlines associated with each axis. The three dashed lines indicate the three threshold matrix elements for the example.

ternary plot. It is possible, and highly likely, that instead of existing as two entirely distinct cohorts, the MEP+ phenotype groups share some overlap, creating a nuanced grey area.

The physiology underlying the subthreshold responses could influence the final threshold matrix position on the ternary plot, hence the upper limb functional outcome. Consider two patients with the same substantial proportion of the subthreshold element, indicating a reduced number of corticomotor neurons, demyelinated axons and subsequent desynchronized output resulting in small, non-modulating MEPs (Byrnes et al., 1999; Rossini & Rossi, 1998; Rossini et al., 2015; Talelli et al., 2006). One scenario involves adequate remyelination processes, proving sufficient for synchronized motor output, transforming subthreshold responses into suprathreshold ones and causing the matrix to shift downwards and leftwards on the ternary plot. Alternatively, if the extent of the damage is too great and remyelination processes are inadequate, synchronized motor output

remains unattainable. In this scenario, the threshold matrix composition might remain unchanged, or subthreshold responses might be converted into subliminal responses, leading to an upward and leftward shift of the ternary plot. It might well be that the starting coordinates of the threshold matrix and the recovery trajectory into different regions of the ternary plot are important. Although a higher proportion of the subthreshold element might not necessarily preclude a patient from achieving a favourable functional outcome with their upper limb, it is plausible that such outcomes will rely on underlying physiological processes influencing the transformation of subthreshold responses to suprathreshold responses.

The TMS protocol used to obtain the threshold matrix can also be used to construct conventional S-R curves. However, as noted above, S-R curves might not accurately reflect corticomotor function in the moderately or severely impaired paretic upper limb early after stroke. Although there are many parameters that can be derived from an S-R curve, such as slope, maximum MEP amplitude (MEPmax), the stimulus intensity needed to obtain 50% of MEPmax (S50), and the estimated threshold from the sigmoid function, none of these adequately captures subthreshold responses, and there is considerable redundancy between them (Kemlin et al., 2019). Furthermore, S-R curve parameters derived from forearm muscles might have lower reliability than intrinsic hand muscles (Carson et al., 2013; Malcolm et al., 2006). Conversely, the area under the entire S-R curve has been shown to be reliable for proximal and distal upper limb muscles (Carson et al., 2013). By adding greater emphasis to subthreshold responses compared with S-R curve parameters, we propose that the threshold matrix framework better quantifies corticomotor function after stroke.

The threshold matrix is proposed as a potential framework for a better understanding of post-stroke motor recovery. The selections of muscles and intensities referred to in this review are merely examples that we have chosen to examine early recovery of the upper limb. The four muscles were chosen for their involvement in wrist and hand movement, which is essential for functional tasks. Other variations are possible. Determining the optimal combination of muscles and intensities remains an open question warranting further investigation. Examining other combinations of muscles, spanning a greater proximal–distal range, or comparing matrices between paretic and non-paretic sides could also be informative. The practicalities around muscle selection would need to be considered carefully if used in a clinical setting. Combining proximal and distal muscles might preclude the use of a single hotspot. Obtaining responses from the non-paretic side could make data collection sessions lengthy and intolerable, particularly at the subacute stage. In summary, muscle

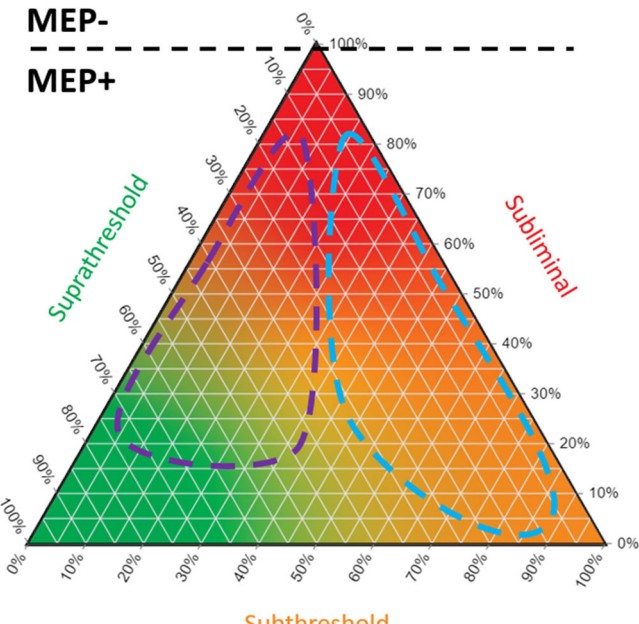

**Figure 4. Overcoming limitations with binarized motor evoked potential status**

The MEP− patients have a 100% subliminal threshold matrix at the top vertex of the ternary plot. Therefore, all regions of the plot below the horizontal black dashed line represent MEP+ patients. The purple dashed ellipse is the hypothesized region of the threshold matrices for MEP+ patients who might achieve a good or excellent upper limb outcome. The light blue dashed ellipse is the hypothesized region of the threshold matrices for MEP+ patients who might not achieve a predicted good upper limb outcome based on MEP status. Note that both the purple and light blue regions are MEP+, indicative of a corticomotor pathway that is at least partly viable. Future research is required to determine whether such regions are non-overlapping and could distinguish between recovery phenotypes. Abbreviation: MEP, motor evoked potential.

and intensity combinations might vary depending on context, such as whether the matrix is designed to gain insight into recovery mechanisms or used clinically for prognostication.

## Conclusion

The use of TMS has significantly advanced our understanding of the neurophysiological mechanisms underlying stroke recovery. The inherent variability of TMS-derived measures poses a challenge for predicting outcomes for individual patients. The present review highlights the need to explore innovative metrics that have the potential to capture more accurately the true extent of damage to the corticomotor pathway and its potential for recovery. The proposed compositional framework represents a promising approach by integrating responses obtained from multiple upper limb muscles and TMS intensities. Future research will determine whether a compositional analysis approach improves prediction accuracy and patient stratification for targeted interventions.

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

## Additional information

### Competing interests

None declared.

### Author contributions

M.J.S. and W.D.B. wrote the paper. Both authors have read and approved the final version of this manuscript and agree to be accountable for all aspects of the work. Both persons designated as authors qualify for authorship, and all those who qualify for authorship are listed.

### Funding

This project was supported by Project Grants from the Health Research Council of New Zealand (20/190, 23/274).

### Acknowledgements

Open access publishing facilitated by The University of Auckland, as part of the Wiley - The University of Auckland agreement via the Council of Australian University Librarians.

### Keywords

human, motor evoked potential, primary motor cortex, stroke, transcranial magnetic stimulation

## Supporting information

Additional supporting information can be found online in the Supporting Information section at the end of the HTML view of the article. Supporting information files available:

**Peer Review History**

