## [Peer Review History · The Journal of Physiology]

Corticomotor pathway function and recovery after stroke: A look back and a way forward

Maxine J Shanks and Winston D Byblow
DOI: 10.1113/JP285562

Corresponding author(s): Winston Byblow (w.byblow@auckland.ac.nz)

The following individual(s) involved in review of this submission have agreed to reveal their identity: Ulrike Hammerbeck (Referee #2); Annapoorna Kuppuswamy (Referee #3)

Review Timeline:

Submission Date:	26-Dec-2023
Editorial Decision:	26-Feb-2024
Revision Received:	13-Mar-2024
Editorial Decision:	01-May-2024
Revision Received:	06-May-2024
Accepted:	15-May-2024

Senior Editor: Laura Bennet

Reviewing Editor: Ricci Hannah

Transaction Report:

Dear Dr Byblow,

Re: JP-TR-2023-285562 "New insights into corticomotor pathway function and recovery after stroke" by Winston D Byblow and Maxine J Shanks

Thank you for submitting your manuscript to The Journal of Physiology. It has been assessed by a Reviewing Editor and by 3 expert referees and we are pleased to tell you that it is potentially acceptable for publication following satisfactory major revision.

REVISION CHECKLIST:

We look forward to receiving your revised submission.

Yours sincerely,

Laura Bennet
Senior Editor
The Journal of Physiology

REQUIRED ITEMS

- Please upload separate high quality figure files via the submission form.

EDITOR COMMENTS

Reviewing Editor:

The reviews from reviewers #1, #2, and #3 paint a nuanced picture of the article, acknowledging both its potential impact and areas needing improvement. Key areas highlighted include clarifying the structure, integrating the review and framework sections more effectively, justifying the proposed method's rationale and superiority.

Reviewer #1 makes a valid observation regarding the overlap with previous work from the same group. Nevertheless, I believe that addressing the concerns raised by each reviewer could address this issue. This includes thorough discussions on limitations and methodological concerns, as well as carefully delineating areas of uncertainty and knowledge gaps.

REFEREE COMMENTS

Referee #1:

This review deals with the use of TMS-derived measures to predict upper limb motor recovery after stroke

Although the manuscript is fairly written, my overall feeling is that there are several shortcomings that limit the impact of this review.

The review is divided into two parts: the first one summarizes our current knowledge on TMS-derived measures in motor recovery after stroke whereas the second one proposes a "novel" threshold matrix framework to predict more accurately the damage to the corticospinal tract and subsequently the potential for recovery.

Unfortunately, the first part of paper is not original at all, there is nothing new since the meta-analysis on this topic proposed by the same group (see McDonnell and Stinear, Brain Stim, 2017). This reference is missing.

From the second part, the description of the "novel" threshold matrix framework in healthy subjects is still not original too, as it is a summary of Shanks et al., EBR, 2023.

Up to me, the only original and novel findings in this review is the application of this matrix to a stroke patient and the discussion about the use of a ternary plot to improve the prediction of recovery. However, I don't understand why it is part of a review. This should be presented separately as a case report or a comment to the main paper (Shanks et al., EBR, 2023). Note that the use of a single ("optimal" in the original paper) hotspot for FDI, ADM, ECR, FCR muscles should be carefully discussed.

Finally, although the lab has an international recognition in this field, there are too many references to their own works.

Referee #2:

Review JP-TR-2023-285562

In this commissioned topical review, New insights into corticomotor pathway function and recovery after stroke, the authors additionally propose a novel framework for examining MEPs to provide a more comprehensive view of corticomotor function early after stroke. The paper makes some interesting points; however, it would benefit from clearer structure to guide the reader and further detail in a number of areas.

Major

The section titled 'TMS after stroke' combines basic TMS method and refers occasionally to changes after stroke. The section would benefit by subheadings to structure the narrative and make reading easier. Potentially: TMS mechanism, TMS delivery/method, single pulse TMS after stroke (potentially with a table of what changes), paired pulse. Alternatively, the subheadings could relate to the underlying physiology and outline the differences seen in stroke.

It could be clearer why the authors propose a framework of stimulating different muscles at both sub and supra-threshold. The case for using subthreshold TMS measures is provided in the review section of the paper. However, although differences in TMS responses in proximal and distal muscles and how this is affected in stroke survivors, are briefly mentioned the rationale for using different muscles is not clearly provided. The case for using suprathreshold responses is not argued at all.

The paper reads as two parts, the review and the framework. It would be useful to integrate the two parts and provide a review of the components of the proposed framework. It would also be useful to include a limitations section here as the number of stimulations required to obtain a matrix like the proposed, with multiple levels of MSO and muscles, could lead to a very high number of stimulations, some of which at very high MSO. It is interesting that all the proposed muscles in the matrix are distal muscles.

Minor points:

Abstract

Awkward long sentence.

Here, we propose a novel framework for examining MEPs, which combines responses from several upper limb muscles across various TMS intensities, to provide a more comprehensive view of corticomotor function early after stroke to better quantify subthreshold as well as suprathreshold MEPs.

The link between the variability of responses and the novel framework could be made clearer by describing the functioning of these first and then stipulating that you are proposing a new framework.

Replace human patients with stroke survivors throughout the paper.

Intro

Page 4 line 96 Could you add Turton et al, 1996

TMS in stroke

Page 5 line 118-119 Please define RMT and AMT abbreviations.

Page 5 line 121 The MEP amplitude used to establish AMT does not provide a reference. If the previous reference Rossini et al, 2015 was used, the method is different to the one described here. 'In active muscles with ongoing activity, MEPs greater than 0.1 mV (100 μ V) are judged to be positive.'

Due to the complexity of using AMT it would be useful to expand a bit more on the standardisation of this response including standardising pre-activation.

P6 line 155 This is a repeat of line 127 - higher RMT and AMT

P6 line 156, Introducing a very different concept of iMEPs would be better at the start of a paragraph. I would remove first line and use reference when you are referring to this in line 127.

P6 line 160 The review looks to propose a framework for subacute stroke but both references for iMEP were in chronic stroke survivors. Please replace with references that investigated sub-acute stroke. There is some work on this- Hammerbeck et al, NNR 2021.

P7 line 194 This has also already been reported with the same references line 135.

P8 Line 205 Reference needed.

P8 Line 226 Repeat

P8 Line 235 You could expand that the most change is likely to be observed in this area with motor recovery after stroke.

P8 Line 236 'All patients who are MEP+ with TMS testing are given a predicted Good outcome.' Awkward sentence, please rephrase.

P9 Line 255 I don't understand what is meant about the Proportional recovery not having the same importance for patients. These are different concepts. 'While proportional recovery does not have the same clinical importance for individual patients as the PREP prediction tools, .'

P10 Line 267 The transition from the intro to the framework is not easy to follow.

P13 line 349-79 This whole section requires references for the proposed neurophysiology. The tone of the second section is very conversational and differs from the first part.

Referee #3:

This paper presents an elegant description of a novel and nuanced framework for assessing integrity of corticospinal pathway after stroke with the aim of increasing the predictive value of such techniques for recovery of motor function. The primary difference between the older version of the prediction protocol and the new proposal is its inclusion of subthreshold responses and using more than one muscle for construction of the prediction matrix. While this certainly increases the granularity with which corticospinal function is interrogated, here are a few areas of discussion that should be considered for quelling the doubts that are likely to arise in a reader's mind

a- This method appears to be a weighted summation of stimulus-response curve of multiple muscles. There is discussion about how S-R curves were previously not useful in predicting recovery. How different is this? Maybe give some examples from other areas where weighted sum has been superior to conventional methods.

b- how is the proposed method of dimensionality reduction superior to other dimensionality reduction methods?

c- What are the minimum number of muscles needed for accurate prediction of recovery? If this is not known, what would be an educated guess?

d- What composition of muscles are required? The example provided are all distal hand muscles. Would inclusion of

forearm and arm muscles add different type of predictive information for functional recovery?

e- What would be the tolerability of such extensive testing of muscles at the acute/subacute phase? I take it not everyone is tolerant of 100% MSO stimulation?

f- Would there be any value in using the unaffected hand responses?

Minor comment

Lines 191-193 - If literature is divided should we not question GABA ergic dysfunction's role? Please add some ref for why GABA dysfunction is recognised.

END OF COMMENTS

Confidential Review

26-Dec-2023

EDITOR COMMENTS

Reviewing Editor:

The reviews from reviewers #1, #2, and #3 paint a nuanced picture of the article, acknowledging both its potential impact and areas needing improvement. Key areas highlighted include clarifying the structure, integrating the review and framework sections more effectively, justifying the proposed method's rationale and superiority.

Reviewer #1 makes a valid observation regarding the overlap with previous work from the same group. Nevertheless, I believe that addressing the concerns raised by each reviewer could address this issue. This includes thorough discussions on limitations and methodological concerns, as well as carefully delineating areas of uncertainty and knowledge gaps.

Referee #1:

This review deals with the use of TMS-derived measures to predict upper limb motor recovery after stroke

Although the manuscript is fairly written, my overall feeling is that there are several shortcomings that limit the impact of this review.

The review is divided into two parts: the first one summarizes our current knowledge on TMS-derived measures in motor recovery after stroke whereas the second one proposes a "novel" threshold matrix framework to predict more accurately the damage to the corticospinal tract and subsequently the potential for recovery.

Unfortunately, the first part of paper is not original at all, there is nothing new since the meta-analysis on this topic proposed by the same group (see McDonnell and Stinear, Brain Stim, 2017). This reference is missing.

Response:

Thank you for noting the omitted reference, which is now included. We have also expanded on new ideas and references about interhemispheric competition, including new meta-analyses that exposes further limitations about interhemispheric competition. (See page 8, line 209-25).

From the second part, the description of the "novel" threshold matrix framework in healthy subjects is still not original too, as it is a summary of Shanks et al., EBR, 2023.

Up to me, the only original and novel findings in this review is the application of this matrix to a stroke patient and the discussion about the use of a ternary plot to improve the prediction of recovery. However, I don't understand why it is part of a review. This should be presented separately as a case report or a comment to the main paper (Shanks et al., EBR, 2023). Note that the use of a single ("optimal" in the original paper) hotspot for FDI, ADM, ECR, FCR

muscles should be carefully discussed.

Response:

Thank you. For topical reviews, the guidelines are to synthesise current research, project forward, discuss where is the field going and raise interesting new questions. Authors are encouraged to be speculative and controversial. We followed these guidelines. The topical review as you mention, has been split into two sections. The application of threshold matrices and ternary plots to a stroke cohort is a novel approach which raises interesting new questions for the field, namely by identifying a way to quantify subthreshold MEPs which are common after stroke but have been largely overlooked up to this point. We have made this structure clearer for the reader by adding the following sentence to the end of the introduction (Page 4 line 101-5):

“This review has two aims. First, it revisits the use of TMS-derived measures used to predict recovery potential and understand post-stroke recovery neurophysiological mechanisms. Second, it introduces a novel TMS-derived framework which may have the potential to expand understanding of these areas.”

The strengths and limitations for using a global hotspot have been covered in our previous article and is referenced accordingly.

Finally, although the lab has an international recognition in this field, there are too many references to their own works.

Response:

Thank you. We have included new references as suggested by reviewers and reduced the number of self-citations wherever possible so as not to generate extensive lists.

Referee #2:

In this commissioned topical review, New insights into corticomotor pathway function and recovery after stroke, the authors additionally propose a novel framework for examining MEPs to provide a more comprehensive view of corticomotor function early after stroke. The paper makes some interesting points; however, it would benefit from clearer structure to guide the reader and further detail in a number of areas.

Major

The section titled 'TMS after stroke' combines basic TMS method and refers occasionally to changes after stroke. The section would benefit by subheadings to structure the narrative and make reading easier. Potentially: TMS mechanism, TMS delivery/method, single pulse TMS after stroke (potentially with a table of what changes), paired pulse. Alternatively, the subheadings could relate to the underlying physiology and outline the differences seen in stroke.

Response:

Thank you for this suggestion. Following your recommendation, we have added four subheadings to the section, and this sentence (Page 5 line 119-21): “The following sections introduce conventional TMS measures, the underlying physiological mechanisms of each measure, and explain how each measure is affected after stroke.”

It could be clearer why the authors propose a framework of stimulating different muscles at both sub and supra-threshold. The case for using subthreshold TMS measures is provided in the review section of the paper. However, although differences in TMS responses in proximal and distal muscles and how this is affected in stroke survivors, are briefly mentioned the rationale for using different muscles is not clearly provided. The case for using suprathreshold responses is not argued at all.

Response:

Thank you. The following sentence has been added (Page 11 line 311-14): “It is important to capture the relative proportions of the subliminal, subthreshold, and suprathreshold elements. One benefit of compositional data analysis methods is that the relationship between the elements is retained while removing the dependency between elements.” Indeed, the extant literature to date has involved protocols which elicit suprathreshold responses, and these are reviewed in detail.

The paper reads as two parts, the review and the framework. It would be useful to integrate the two parts and provide a review of the components of the proposed framework. It would also be useful to include a limitations section here as the number of stimulations required to obtain a matrix like the proposed, with multiple levels of MSO and muscles, could lead to a very high number of stimulations, some of which at very high MSO. It is interesting that all the proposed muscles in the matrix are distal muscles.

Response:

Thank you. The guidelines for topical review are to synthesise current research and project forward and discuss where is the field going and raise interesting new questions. Authors are encouraged to be speculative and controversial. In order to follow these guidelines, the topical review as you mention, has been split into two sections. This structure is now more clearly described as mentioned above.

(Page 4 line 101-5):

“This review has two aims. First, it revisits the use of TMS-derived measures used to predict recovery potential and understand post-stroke recovery neurophysiological mechanisms. Second, it introduces a novel TMS-derived framework which may have the potential to expand understanding of these areas.”

Additionally, we have added a limitations section to the end of the review (Page 15 line 416-29).

Minor points:

Abstract

Awkward long sentence.

Here, we propose a novel framework for examining MEPs, which combines responses from several upper limb muscles across various TMS intensities, to provide a more comprehensive view of corticomotor function early after stroke to better quantify subthreshold as well as suprathreshold MEPs.

Response:

Thank you for your comment. The sentence had been replaced with two sentences and now reads (Page 2 line 50-4): “Here, we propose a novel compositional framework to examine MEPs across several upper limb muscles within a threshold matrix. The matrix may provide a more comprehensive view of corticomotor function and recovery after stroke by quantifying the evolution of subthreshold and suprathreshold MEPs through compositional analyses.”

The link between the variability of responses and the novel framework could be made clearer by describing the functioning of these first and then stipulating that you are proposing a new framework.

Response:

It is unclear what variability and functioning the reviewer is referring to here.

Replace human patients with stroke survivors throughout the paper.

Response:

Thank you for your suggestion. While we agree with the suggestion to avoid patients when participants or survivors would be more accurate, that is simply not the case in this context. The use of “patients” is appropriate in contexts where MEP status is used to make predictions about upper limb outcome for individual patients in a hospital setting, which is how the prediction algorithms are used. Therefore “patients” has been retained throughout the manuscript for internal consistency and consistency with literature in this domain.

Intro

Page 4 line 96 Could you add Turton et al, 1996

Response:

Thank you for the suggestion. However, the point being raised in Page 4 line 96 is that in recent years MEP status (i.e., the presence or absence of a MEP) has been identified as a biomarker for upper limb recovery after stroke. While we acknowledge there has been substantial literature published investigating motor evoked potential properties after stroke (e.g., Turton et al., 1996), this literature does not discuss or propose the use of MEP status as a biomarker after stroke. Therefore, this reference does not support the statement we are making. We refer to the seminal findings of Turton et al elsewhere, as appropriate.

TMS in stroke

Page 5 line 118-119 Please define RMT and AMT abbreviations.

Response:

Thank you, the definitions of the abbreviations have been added. Page 5 line 124-6 now reads:

“Measures of motor threshold reflect the ease at which the corticomotor pathway is activated by TMS. They can be determined with either the muscle at rest (resting motor threshold, RMT) or in a pre-activated state (active motor threshold, AMT).”

Page 5 line 121 The MEP amplitude used to establish AMT does not provide a reference. If the previous reference Rossini et al, 2015 was used, the method is different to the one described here. 'In active muscles with ongoing activity, MEPs greater than 0.1 mV (100 μ V) are judged to be positive.'

Response:

Thank you, we have added the more recent reference of Vucic et al., 2023, which reflect the current IFCN guidelines.

Due to the complexity of using AMT it would be useful to expand a bit more on the standardisation of this response including standardising pre-activation.

Response:

Thank you. The following sentence has been added (Page 5 line 129-32). “Numerous technical and physiological factors should be kept consistent when determining motor threshold, such as coil position, the level of background activity in the target muscle and environmental noise (Rossini et al., 2015).

P6 line 155 This is a repeat of line 127 - higher RMT and AMT

P6 line 156, Introducing a very different concept of iMEPs would be better at the start of a paragraph. I would remove first line and use reference when you are referring to this in line 127.

Response:

Thank you, the first sentence of the paragraph has been removed and now the iMEP section (Page 6 line 165-7) begins with “After stroke, MEPs in the paretic upper limb may also be present when TMS is applied to the contralesional (ipsilateral) motor cortex, usually at high intensities (Turton et al., 1996).”

P6 line 160 The review looks to propose a framework for subacute stroke but both references for iMEP were in chronic stroke survivors. Please replace with references that investigated sub-acute stroke. There is some work on this- Hammerbeck et al, NNR 2021.

Response:

Thank you for your comment. The following references to studies investigating iMEPs at the subacute stage after stroke have been added (Page 7 line 179-80). Bastings et al., 1997; Alagona et al., 2001; Hammerbeck et al., 2021.

P7 line 194 This has also already been reported with the same references line 135.

Response:

Thank you. The repeated sentence has been removed and the paragraph has been restructured (Page 8 line 209-25).

P8 Line 205 Reference needed.

Response:

Thank you. The Rossini et al., 2015 reference has been added to support this sentence.

P8 Line 226 Repeat

Response:

Thank you, the start of the paragraph now reads (Page 9 line 248-50) “Resting motor threshold and MEP status are conventional, robust metrics used at the subacute stage after stroke. However, they are not without limitations. The conventionally accepted definition for RMT is based on both amplitude and persistence criteria (Rossini *et al.*, 2015).”

P8 Line 235 You could expand that the most change is likely to be observed in this area with motor recovery after stroke.

Response:

Thank you for this suggestion. See Page 9 line 259-60.

P8 Line 236 'All patients who are MEP+ with TMS testing are given a predicted Good outcome.' Awkward sentence, please rephrase.

Response:

Thank you. The sentence has been re-worded (Page 9 line 261-7): “For example, the PREP2 prediction tool can be used to predict upper limb outcomes for individual patients within one week of stroke. Patients with upper limb weakness receive TMS to determine their MEP status. If they are MEP+ they are predicted to achieve a Good functional outcome for their hand and arm at 12 weeks, even though they have little or no voluntary movement at the time of testing. Most of these patients meet or exceed their outcome. However, 20% fall short of their predicted outcome (Stinear *et al.*, 2017b).”

P9 Line 255 I don't understand what is meant about the Proportional recovery not having the same importance for patients. These are different concepts. 'While proportional recovery does not have the same clinical importance for individual patients as the PREP prediction tools, '

Response:

Thank you. The important link between these two ideas is now better explained (Page 10 line 273-86).

P10 Line 267 The transition from the intro to the framework is not easy to follow.

Response:

Thank you. As mentioned in a response to an earlier comment we have made the structure of the review clearer for the reader by adding the following sentence to the end of the introduction (Page 4 line 101-5): “This review has two aims. First, it revisits the use of TMS-derived measures used to predict recovery potential and understand post-stroke recovery neurophysiological mechanisms. Second, it introduces a novel TMS-derived framework which may have the potential to expand understanding of these areas.”

P13 line 349-79 This whole section requires references for the proposed neurophysiology. The tone of the second section is very conversational and differs from the first part.

Response:

Thank you. The following references for the proposed neurophysiology have been added (Page 14 line 388-9): Rossini & Rossi, 1998; Byrnes *et al.*, 1999; Talelli *et al.*, 2006; Rossini *et al.*, 2015. However, the majority of this section reflects our working hypothesis about how the threshold matrix framework might apply in stroke recovery and references for some statements cannot be provided.

Referee #3:

This paper presents an elegant description of a novel and nuanced framework for assessing integrity of corticospinal pathway after stroke with the aim of increasing the predictive value of such techniques for recovery of motor function. The primary difference between the older version of the prediction protocol and the new proposal is its inclusion of subthreshold responses and using more than one muscle for construction of the prediction matrix. While this certainly increases the granularity with which corticospinal function is interrogated, here are a few areas of discussion that should be considered for quelling the doubts that are likely to arise in a reader's mind

a- This method appears to be a weighted summation of stimulus-response curve of multiple muscles. There is discussion about how S-R curves were previously not useful in predicting recovery. How different is this? Maybe give some examples from other areas where weighted sum has been superior to conventional methods.

Response:

Thank you. The new section has been added to explain similarities and differences with S-R curve parameters Page 14 lines 405-15.

b- how is the proposed method of dimensionality reduction superior to other dimensionality reduction methods?

Response:

Thank you for this thought-provoking question. Given the redundancy noted between multiple curve parameters in the study of Kemlin et al (2019) a PCA based on multiple parameters could also be considered. It remains to be determined. As noted above, the main benefit of a threshold matrix framework is to capture the subthreshold responses, which are common after stroke and frequently overlooked, and not adequately represented by S-R curve parameterisation, which therefore would not lend itself to PCA.

c- What are the minimum number of muscles needed for accurate prediction of recovery? If this is not known, what would be an educated guess?

d- What composition of muscles are required? The example provided are all distal hand muscles. Would inclusion of forearm and arm muscles add different type of predictive information for functional recovery?

e- What would be the tolerability of such extensive testing of muscles at the acute/subacute phase? I take it not everyone is tolerant of 100% MSO stimulation?

f- Would there be any value in using the unaffected hand responses?

Response:

Thank you. Indeed, the threshold matrix concept and framework is in its infancy, hence is discussed in a speculative, hypothesis generating way in this topical review. To address items c-f we have added additional information on Page 15 line 416-29).

Minor comment

Lines 191-193 - If literature is divided should we not question GABA ergic dysfunction's role? Please add some ref for why GABA dysfunction is recognised.

Response:

Thank you. We have elaborated on this point added more detail to provide more explicit links between ideas. See Page 7 line 183-208.

Dear Professor Byblow,

Re: JP-TR-2024-285562R1 "New insights into corticomotor pathway function and recovery after stroke" by Winston D Byblow and Maxine J Shanks

Thank you for submitting your manuscript to The Journal of Physiology. It has been assessed by a Reviewing Editor and by 2 expert referees and we are pleased to tell you that it is acceptable for publication following satisfactory revision.

ABSTRACT FIGURES: Authors may use The Journal's premium BioRender account to create/redraw their Abstract Figures (and any other suitable schematic figure). Information on how to access this account is here: <https://physoc.onlinelibrary.wiley.com/journal/14697793/biorender-access>.

REVISION CHECKLIST: Upload a full Response to Referees file. To create your 'Response to Referees' copy all the reports, including any comments from the Senior and Reviewing Editors, into a Microsoft Word, or similar, file and respond to each point, using font or background colour to distinguish comments and responses and upload as the required file type.

We look forward to receiving your revised submission.

Yours sincerely,

EDITOR COMMENTS

I'm pleased to inform you that two of the three original reviewers have given positive feedback on the revised manuscript, highlighting its potential influence. Despite one reviewer not reassessing the manuscript, considering all viewpoints, I recommend provisional acceptance. Reviewer #2 has a few minor comments to address. In particular, they suggest rewording the title to better reflect the forward-looking perspective contained within the article.

REFEREE COMMENTS

Referee #2:

The authors have addressed the concerns and suggestions.

However, the title is misleading as the topical review consists, as the authors point out, of a review and an introduction of a novel framework that MAY have the potential to expand understanding ...

Therefore could this sentiment be reflected in the title, rather than the statement of New insights....

P22 Line 411 The threshold matrix analysis can be conceptualised as an area under S-R curve measure across multiple muscles based on the reliable measure of RMT.

This sentence is difficult to follow.

Referee #3:

Thank you for responding to my queries about how this method is better than others and what more needs to be done to make it a possible clinical tool. I have no further comments.

END OF COMMENTS

1st Confidential Review

13-Mar-2024

Response to Referees – R2

EDITOR COMMENTS

I'm pleased to inform you that two of the three original reviewers have given positive feedback on the revised manuscript, highlighting its potential influence. Despite one reviewer not reassessing the manuscript, considering all viewpoints, I recommend provisional acceptance. Reviewer #2 has a few minor comments to address. In particular, they suggest rewording the title to better reflect the forward-looking perspective contained within the article.

Referee #2:

The authors have addressed the concerns and suggestions.

However, the title is misleading as the topical review consists, as the authors point out, of a review and an introduction of a novel framework that MAY have the potential to expand understanding ...

Therefore could this sentiment be reflected in the title, rather than the statement of New insights....

Thank you, we have changed the title accordingly: “Corticomotor pathway function and recovery after stroke: a look back and a way forward”

P22 Line 411 The threshold matrix analysis can be conceptualised as an area under S-R curve measure across multiple muscles based on the reliable measure of RMT.

This sentence is difficult to follow.

Thank you, we apologise for the confusion and have removed the offending sentence.

Referee #3:

Thank you for responding to my queries about how this method is better than others and what more needs to be done to make it a possible clinical tool. I have no further comments.

Dear Professor Byblow,

Re: JP-TR-2024-285562R2 "Corticomotor pathway function and recovery after stroke: A look back and a way forward" by Maxine J Shanks and Winston D Byblow

We are pleased to tell you that your paper has been accepted for publication in The Journal of Physiology.

Authors should note that it is too late at this point to offer corrections prior to proofing. Major corrections at proof stage, such as changes to figures, will be referred to the Editors for approval before they can be incorporated. Only minor changes, such as to style and consistency, should be made at proof stage. Changes that need to be made after proof stage will usually require a formal correction notice.

Yours sincerely,

Laura Bennet
Senior Editor
The Journal of Physiology

P.S. - You can help your research get the attention it deserves! Check out Wiley's free Promotion Guide for best-practice recommendations for promoting your work at www.wileyauthors.com/eoo/guide. You can learn more about Wiley Editing Services which offers professional video, design, and writing services to create shareable video abstracts, infographics, conference posters, lay summaries, and research news stories for your research at www.wileyauthors.com/eoo/promotion.

IMPORTANT NOTICE ABOUT OPEN ACCESS: To assist authors whose funding agencies mandate public access to published research findings sooner than 12 months after publication, The Journal of Physiology allows authors to pay an Open Access (OA) fee to have their papers made freely available immediately on publication.

You can check if your funder or institution has a Wiley Open Access Account here: <https://authorservices.wiley.com/author-resources/Journal-Authors/licensing-and-open-access/open-access/author-compliance-tool.html>.

EDITOR COMMENTS

Reviewing Editor:

Thank you for addressing the final comments. I am happy to recommend that the article be accepted for publication.